# Integration of locomotion and auditory signals in the mouse inferior colliculus

**Yoonsun Yang[1,2], Joonyeol Lee[1,3], Gunsoo Kim[1,3]***

[1]Center for Neuroscience Imaging Research (CNIR), Institute for Basic Science (IBS), Suwon, Republic of Korea; [2]Department of Physiology, School of Medicine, Sungkyunkwan University, Suwon, Republic of Korea; [3]Department of Biomedical Engineering, Sungkyunkwan University, Suwon, Republic of Korea

**Abstract** The inferior colliculus (IC) is the major midbrain auditory integration center, where virtually all ascending auditory inputs converge. Although the IC has been extensively studied for sound processing, little is known about the neural activity of the IC in moving subjects, as frequently happens in natural hearing conditions. Here, by recording neural activity in walking mice, we show that the activity of IC neurons is strongly modulated by locomotion, even in the absence of sound stimuli. Similar modulation was also found in hearing-impaired mice, demonstrating that IC neurons receive non-auditory, locomotion-related neural signals. Sound-evoked activity was attenuated during locomotion, and this attenuation increased frequency selectivity across the neuronal population, while maintaining preferred frequencies. Our results suggest that during behavior, integrating movement-related and auditory information is an essential aspect of sound processing in the IC.

## Introduction

The inferior colliculus (IC) is the major auditory integration center in the midbrain, where virtually all ascending inputs from the auditory brainstem and descending cortical inputs converge (*Adams, 1979*; *Adams, 1980*; *Malmierca, 2004*; *Winer and Schreiner, 2005*). The IC plays a critical role in representing spectrotemporal features and communication sounds (*Egorova et al., 2001*; *Escabí and Schreiner, 2002*; *Lesica and Grothe, 2008*; *Woolley and Portfors, 2013*), and localizing sound sources (*Bock and Webster, 1974*; *Schnupp and King, 1997*; *Lesica et al., 2010*; *Xiong et al., 2013*; *Ono and Oliver, 2014*). While auditory response properties have been extensively studied, most IC neural activities have been recorded in stationary subjects, and little is known about IC activity in subjects engaged in locomotion.

The central nucleus of the IC predominantly receives auditory inputs. However, the shell region of the IC (the lateral and dorsal cortex) receives non-auditory projections such as somatosensory inputs (*Cooper and Young, 1976*; *Morest and Oliver, 1984*; *Coleman and Clerici, 1987*; *Lesicko et al., 2016*) and performs multi-sensory integration (*Aitkin et al., 1978*; *Aitkin et al., 1981*; *Jain and Shore, 2006*). This region has also been implicated in generating sound-driven behavior by projecting to motor-related regions (*Huffman and Henson, 1990*; *Xiong et al., 2015*). Although this multi-modal integration may serve a range of functions (*Gruters and Groh, 2012*), modulation of the IC activity during vocalization (*Schuller, 1979*; *Tammer et al., 2004*) or eye movements (*Groh et al., 2001*; *Porter et al., 2006*; *Porter et al., 2007*) suggests that the main role of non-auditory inputs is to provide motor-related information.

In order to accurately detect and localize sounds during movement, it is hypothesized that the auditory system distinguishes self-generated sounds from external ones (*Poulet and Hedwig, 2002*; *Rummell et al., 2016*; *Schneider et al., 2018*) and integrates movement-related signals with auditory information. Recent studies in behaving mice show that movements such as locomotion strongly

*For correspondence:
kgunsoo@skku.edu

Competing interests: The authors declare that no competing interests exist.

modulate neural activity in the mouse primary auditory cortex (A1) (*Schneider et al., 2014*; *Zhou et al., 2014*; *McGinley et al., 2015*; *Bigelow et al., 2019*). While A1 neurons receive movement-related signals from sources outside the auditory pathway (*Schneider et al., 2014*; *Nelson et al., 2013*; *Nelson and Mooney, 2016*; *Reimer et al., 2016*), movement-related modulation is also found in subcortical auditory centers. In the medial geniculate body (MGB), for instance, sound-evoked activity is attenuated during locomotion (*Williamson et al., 2015*; *McGinley et al., 2015*). Motor-related modulation is also found in the auditory brainstem during vocalization, licking, and pinna orientation (*Suga and Schlegel, 1972*; *Schuller, 1979*; *Kanold and Young, 2001*; *Singla et al., 2017*). These subcortical modulation studies suggest that the integration loci for movement-related and auditory information are spread out along the auditory pathway.

Here, we investigated locomotion-related modulation of the IC neural activity in head-fixed mice. Our data show that both spontaneous and sound-evoked activities of IC neurons are strongly modulated during locomotion. Both excitation and suppression were found in spontaneous activity, whereas sound-evoked activity was mostly attenuated, decreasing frequency tuning width. Our results suggest that auditory midbrain neurons integrate movement-related information, which may be important for auditory processing during movement and acoustically guided behavior.

## Results

### Spontaneous neural activity of IC neurons is modulated during locomotion

We made extracellular recordings of spiking neural activity of IC neurons in awake head-fixed mice, placed on a passive treadmill. This preparation enabled us to observe the IC neural activity during locomotion. When the firing rates between stationary and walking periods were compared, in the absence of sound stimulus presentation, we found that the IC neuron firing rates could be strongly modulated during locomotion. Among the 96 recorded IC neurons, 51 (*Figure 1A*; 53%) significantly increased their firing during locomotion, whereas 22 (*Figure 1B*; 23%) decreased their firing (*Figure 1C*). In 23 neurons (24%), firing rates did not significantly differ between the stationary and walking periods. Neurons that increased firing showed a positive correlation between the firing rate and walking speed, whereas those that decreased firing showed a negative correlation (*Figure 1D*).

Subdivisions of the IC differ in cytoarchitecture and projection patterns (*Morest and Oliver, 1984*), with the lateral cortex, for example, known to receive somatosensory inputs (*Lesicko et al., 2016*). Therefore, we evaluated whether neurons that showed locomotion-related modulation are clustered in the shell region. Anatomical reconstruction of the recording locations did not show any clustering in terms of modulation or the direction of modulation (*Figure 1E*). Instead, all three types of modulations – increased (red), decreased (blue), or no change (black) - were found across the IC with no clear pattern.

When mice walk on a treadmill, their movements generate low-intensity sounds. These sounds may modulate IC neuron activity. We examined this contribution of auditory reafference by playing back recorded walking sounds at a level similar to the recording (30 dB, n = 25; *Figure 1—figure supplement 1*). We also presented the walking sounds at a substantially higher level (40 dB) to examine the variation in neural response. Playbacks evoked neural responses, but the response magnitudes did not significantly correlate with those induced by locomotion (*Figure 1—figure supplement 1D–G*). This suggests that although some of the observed modulation may be induced by walking sounds, it is unlikely that the modulation is mainly due to auditory reafference.

### Neural modulation precedes locomotion onset

If the firing rate change starts before locomotion onset, it indicates that modulation is not simply due to auditory reafference (*Schneider et al., 2014*). We determined the relative timing of modulation by performing locomotion onset-triggered firing-rate averaging. We found that firing rate changes can begin well before the onset of locomotion (*Figure 2A and B*). In most of the 30 neurons that yielded a significant onset-triggered average (see Materials and methods), the onset of firing-rate modulation preceded the locomotion onset (*Figure 2C*, n = 30, median latency = −107 msec). These negative latencies were observed regardless of the direction of the modulation (*Figure 2D*; positive modulation: red bar, n = 22, –118 msec; negative modulation: blue bar, n = 8, –60 msec;

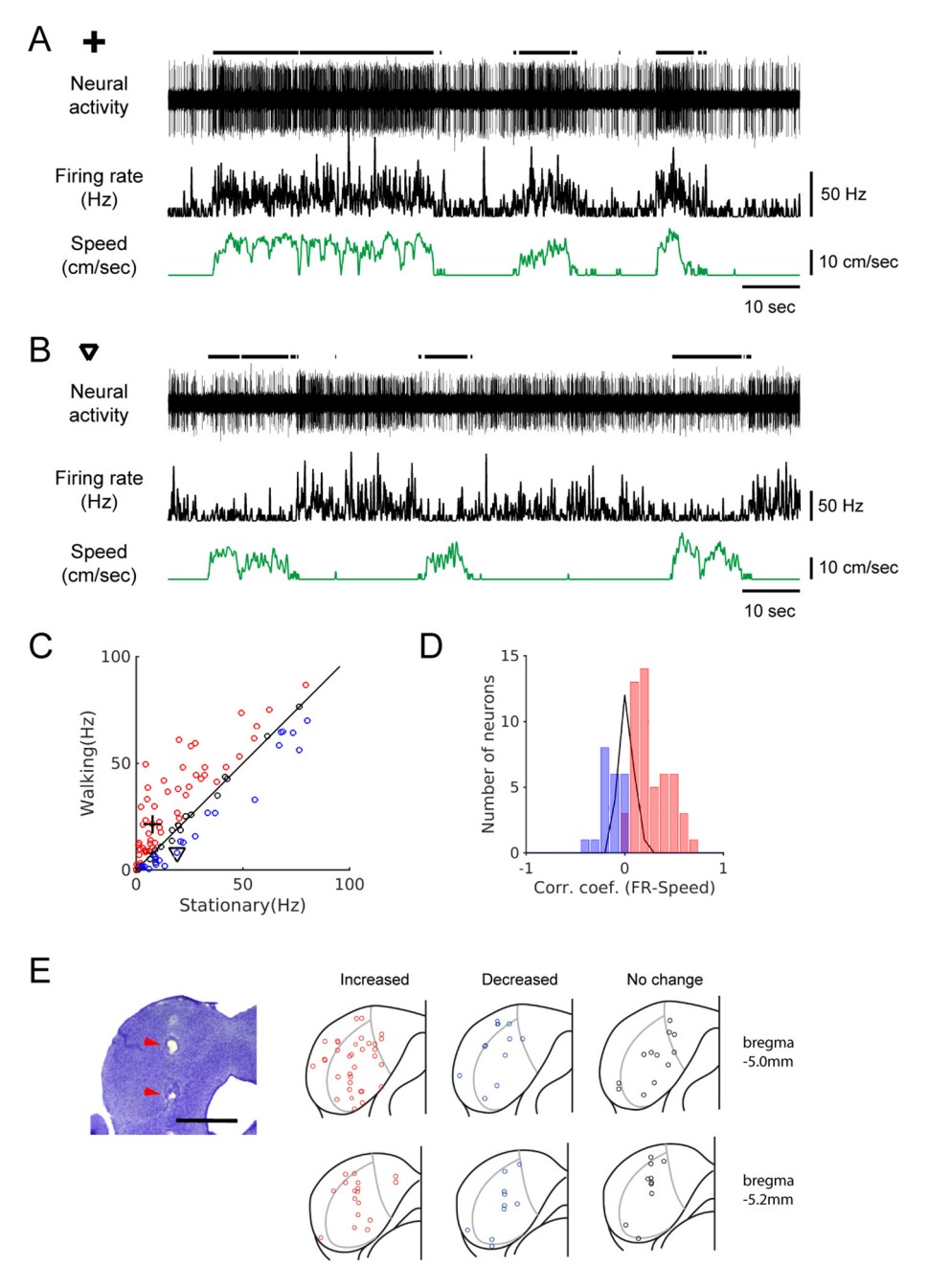

**Figure 1.** Spontaneous activity of IC neurons is modulated by locomotion. (A-B) Example recordings of spontaneous activity in IC neurons. During walking periods, the neuron in A increased firing (from 7.6 Hz to 21.6 Hz), whereas the neuron in B decreased firing (from 19.1 Hz to 8.2 Hz). In both cases, the smoothed firing rates (black middle traces) exhibit significant correlations with the speed of the treadmill (green traces) (A: r = 0.59; B: r = −0.24). Thick black lines above the neural records indicate walking periods. (C) Population plot comparing average spontaneous firing rates between stationary and walking conditions (n = 96 neurons). Red circles: increased, blue: decreased, black: no significant change in firing. Values for the example neurons in A (cross) and B (triangle) are also shown. (D) Histogram of correlation coefficients between smoothed firing rate and speed (color code as in C). (E) Photomicrograph of a Nissl section with lesion sites marked with red arrow heads (scale bar = 1 mm), and reconstructed recording locations are shown in two transverse sections (5.0 and 5.2 mm posterior to the bregma, respectively). Color code as in C.

The online version of this article includes the following source data and figure supplement(s) for figure 1:

*Figure 1 continued on next page*

*Figure 1 continued*

**Source data 1.** Source data for spontaneous activity modulation, and walking sound recording and playback.
**Figure supplement 1.** Recording of walking sounds and playback responses.

Wilcoxon rank sum test, p=0.25). This demonstrates that neural modulation can begin before any detectable movement, and is, therefore, not simply an auditory response to walking sounds.

## Locomotion modulates spontaneous activity in hearing-impaired mice

To further substantiate that non-auditory neural signals modulate IC activity during locomotion, we performed bilateral deafening by removing the middle ear ossicles and applying an ototoxin (kanamycin) to the cochlea (see Materials and methods; n = 4 mice). Systematic mapping of multi-unit responses to broadband noise across the IC showed that this deafening raised the hearing threshold to ≥70 dB, at least 40 dB higher than the threshold in normal mice (*Figure 3A and B*). We reasoned that in these mice, it would be highly unlikely that low-intensity walking sounds (~30 dB, *Figure 1—*

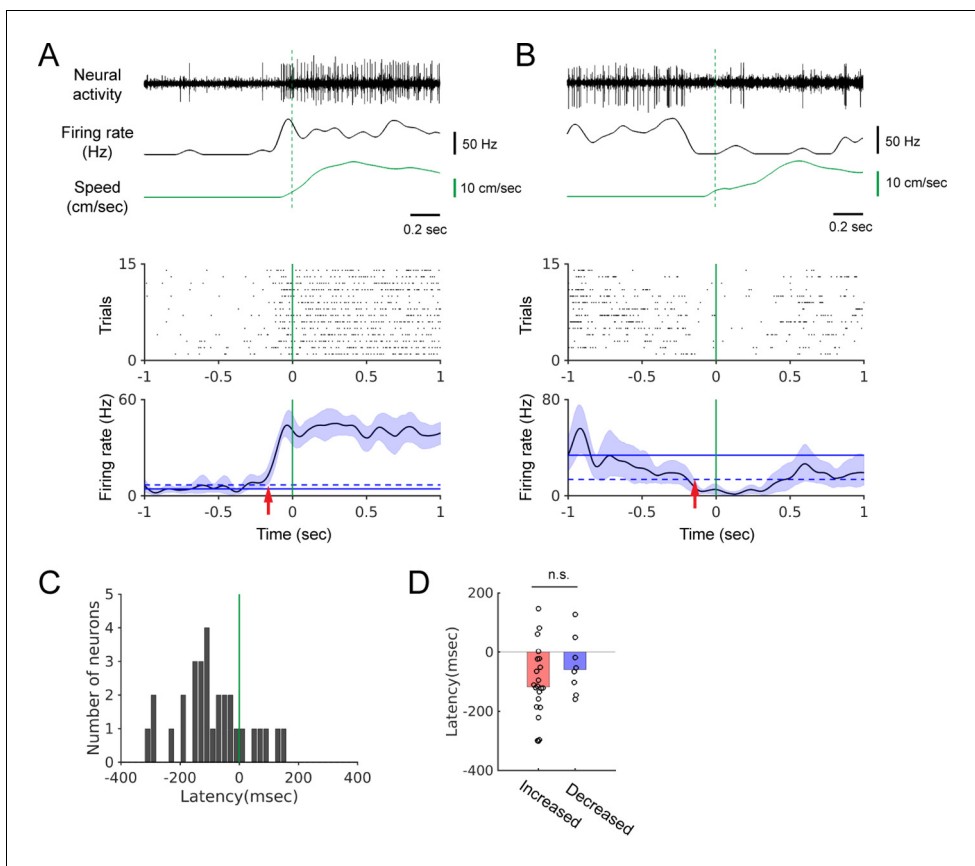

**Figure 2.** The onset of firing rate modulation precedes locomotion onset. (A-B) Top: Example neural recordings, corresponding smoothed firing rates, and walking speed. Bottom: Spike rasters aligned at the onset of walking and the locomotion onset-triggered averages of firing rates. Blue shading indicates the 95% confidence interval of the average firing rates. Horizontal solid and dashed lines show the average spontaneous firing rate and the 2x standard deviation above (A) or below (B) the average, respectively. Vertical green lines indicate locomotion onset. Red arrows indicate modulation onset. (C) Histogram of the neural modulation latencies relative to locomotion onset (n = 30). Time zero indicates locomotion onset. (D) Modulation latencies for neurons with increased (red bar, n = 22) and decreased (blue bar, n = 8) firing during locomotion (Wilcoxon rank sum test, p=0.25). Bar graphs show the median latencies.

The online version of this article includes the following source data for figure 2:

**Source data 1.** Source data for locomotion onset-triggered firing rate averaging and neural modulation latencies.

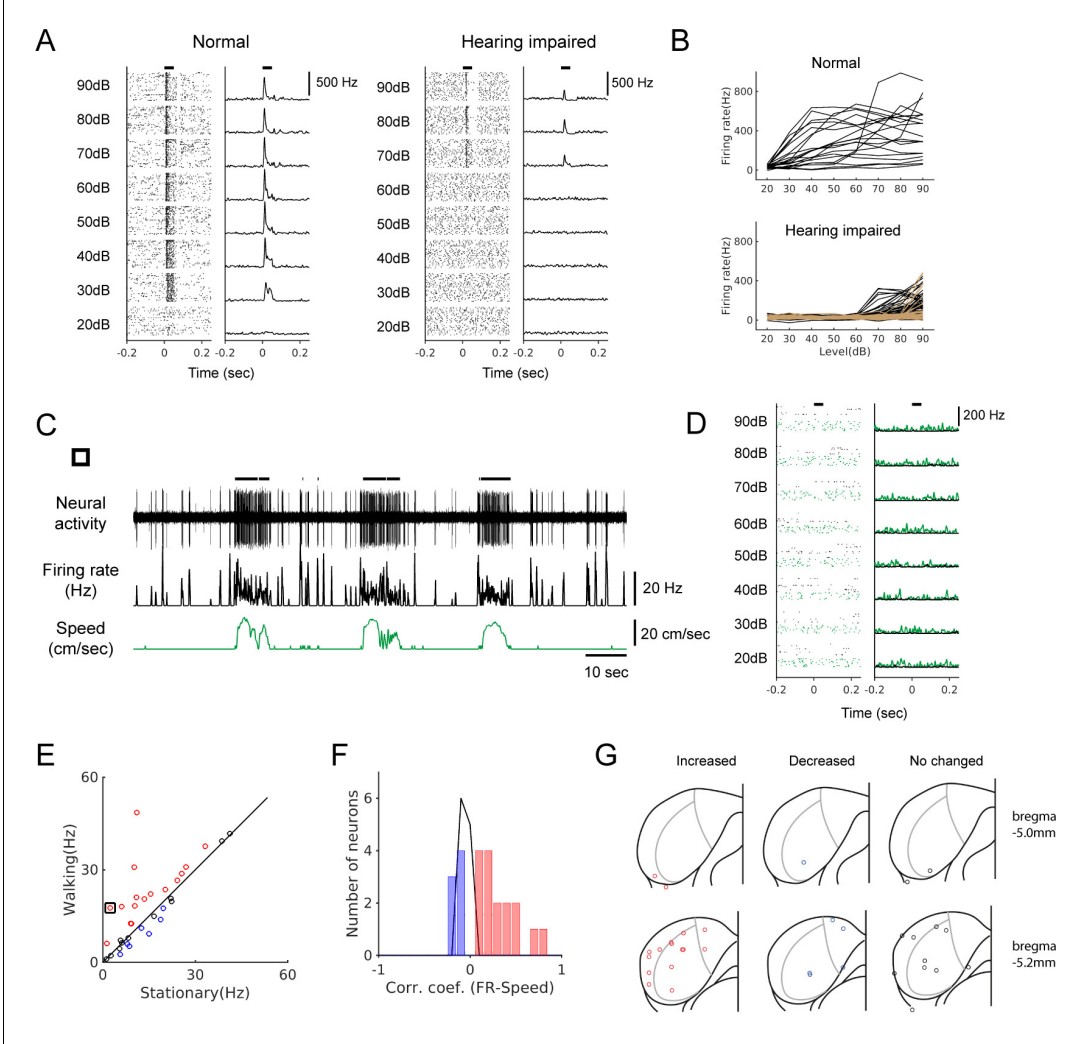

**Figure 3.** Locomotion modulates spontaneous activity in hearing-impaired mice. (**A**) Raster plots (30 trials shown for each level) and PSTHs of multi-unit responses to broadband sound (2–64 kHz, 50 msec duration) in the example sites from normal (left) and hearing-impaired (right) mice. Horizontal bars at the top denote the period of stimulus presentation. (**B**) Rate-level functions from multi-unit sites from a normal mouse (top, 20 sites) and two hearing-impaired mice (bottom, 40 sites (black) and 57 sites (brown); colors represent different mice, and curves from only two mice are shown for clarity). (**C**) Example neuron from a hearing-impaired mouse that showed a robust modulation in firing rate during locomotion. Thick black lines above the neural record indicate walking periods. (**D**) The neuron in C did not show any auditory response to broadband noise (black: stationary, green: locomotion). In the raster plots, 30 stationary trials (black dots) and 8–10 walking trials (green dots) are shown for each level. (**E**) Scatter plot comparing the average spontaneous firing rates between stationary and walking conditions in hearing-impaired mice (n = 34). Red: increased; blue: decreased; black: no significant change. Black square indicates the neuron in C. (**F**) Histogram of correlation coefficients (color code as in E). (**G**) Reconstructed recording locations as in *Figure 1*. Color code as in E.

The online version of this article includes the following source data and figure supplement(s) for figure 3:

**Source data 1.** Source data for the effects of deafening, locomotion-induced modulation in hearing-impaired mice, and comparison of modulation between normal and hearing-impaired mice.

**Figure supplement 1.** Modulation of spontaneous activity is similar between normal and hearing-impaired mice.

*figure supplement 1*) would evoke neural responses. As demonstrated in *Figure 3C*, a neuron from a hearing-impaired mouse showed a robust increase in firing during locomotion, while the same neuron did not show any discernible response to broadband sound stimuli (*Figure 3D*). We observed both increases and decreases in neural firing during locomotion in hearing-impaired mice, as in

normal-hearing mice (n = 34; *Figure 3E and F*). We confirmed anatomically that our recording sites in the hearing-impaired mice were in the IC, and as in normal mice, modulated neurons were found across the IC without any spatial pattern (*Figure 3G*).

Deafening may induce plasticity in the IC during the period before neural recording (8.2 ± 1.1 days; *Chambers et al., 2016*). However, not only that locomotion-modulated neurons were found in hearing-impaired mice, but also the modulation was similar to that in normal mice in terms of the types, magnitudes, and latencies (*Figure 3—figure supplement 1*). Although we cannot exclude the influence of deafening-induced plasticity, the existence of locomotion-modulated neurons in hearing-impaired mice with similar characteristics provides strong evidence that non-auditory signals modulate the activity of IC neurons during locomotion.

## Sound-evoked activity of IC neurons is attenuated during locomotion

Attenuation of sound-evoked activity during locomotion has been shown in the auditory thalamus and cortex, but sources of subcortical attenuation remain unclear (*Schneider et al., 2014*; *Zhou et al., 2014*; *Williamson et al., 2015*; *McGinley et al., 2015*). We investigated whether the sound-evoked IC activity is also modulated during locomotion by presenting pure tones of different frequencies at 70 dB (*Figure 4*).

We found that most IC neurons with excitatory tone-evoked responses showed significant attenuation during locomotion (72%, n = 47/65; *Figure 4A–C and G*). Percent changes in evoked response across the population showed a significant shift toward negative values (*Figure 4H*, n = 65; one sample t-test, p=1.4×10$^{-10}$) with a mean change of −36 ± 5%. Percent attenuation was greater in neurons with relatively weaker responses (below the median) than in those with stronger responses (above the median) (*Figure 4I*, left; stronger: −17 ± 4%, n = 33; weaker: −56 ± 7%, n = 32; t-test, p=0.0175). For individual neurons, average attenuations at non-best frequencies were greater than those at a neuron's best frequency (*Figure 4I*, right; n = 43, −27 ± 5% (BF) vs. −48 ± 7% (non-BF); paired t-test, p=6.98×10$^{-4}$). The greater attenuation of weaker responses within and across neurons may improve the signal-to-noise ratio across the neuronal population during locomotion.

To examine whether the degree of attenuation depends on neuronal response properties, we divided IC neurons into two response types: those that showed responses only at the onset of a stimulus (e.g., *Figure 4C*; n = 30), and those that showed sustained responses throughout the stimulus (e.g., *Figure 4B*; n = 35). We did not find significant differences between the two types (*Figure 4J*; −45 ± 6% vs. −29 ± 7%; t-test, p=0.103). The degree of attenuation also did not correlate with best frequencies, indicating the attenuation was global, rather than frequency-specific (*Figure 4K*, n = 65; r = −0.0031, p=0.98).

Attenuation of evoked activity indicates a decrease in response gain, and this may affect the frequency selectivity of a neuron. To examine this possibility, we constructed frequency tuning curves and quantified tuning widths as the spread around the centroid (*Escabí et al., 2007*; *Ono et al., 2017*; *Figure 4D–F,L,M*). In the stationary condition, the tuning widths ranged from 0.5 to 5 octaves with a mean of 2.2 octaves (n = 65). During locomotion, there was a significant decrease in the tuning widths across the population (*Figure 4L–M*, n = 60, excluding five neurons with tuning width of 0 during walking due to near-complete suppression; one sample Wilcoxon signed rank test, p=3.0×10$^{-7}$; mean change = −0.29 octaves). In contrast, best frequencies did not change during locomotion in the vast majority of the neurons (n = 60, *Figure 4N*). Together, these results demonstrate sound-evoked activity in the IC is attenuated during locomotion, and this attenuation significantly sharpens frequency tuning across the population.

We next evaluated whether there is a relationship between the modulation of spontaneous and sound-evoked activity, which might indicate shared neural mechanisms. Attenuation of evoked activity occurred regardless of spontaneous activity modulation types, but the degree of attenuation correlated with the suppression of spontaneous activity (*Figure 4—figure supplement 1*). Therefore, locomotion-induced suppression of spontaneous and evoked activity may share common neural mechanisms, distinct from excitation. In addition, spontaneous modulation differed depending on the evoked response patterns. In the sustained type, positive modulation was the most common, whereas in the suppressed type, negative modulation was the most common (*Figure 4—figure supplement 2*). These correlations suggest an overlap between modulatory and sound-driven inputs, but this overlap is partial because the suppressed type can also be excited by locomotion.

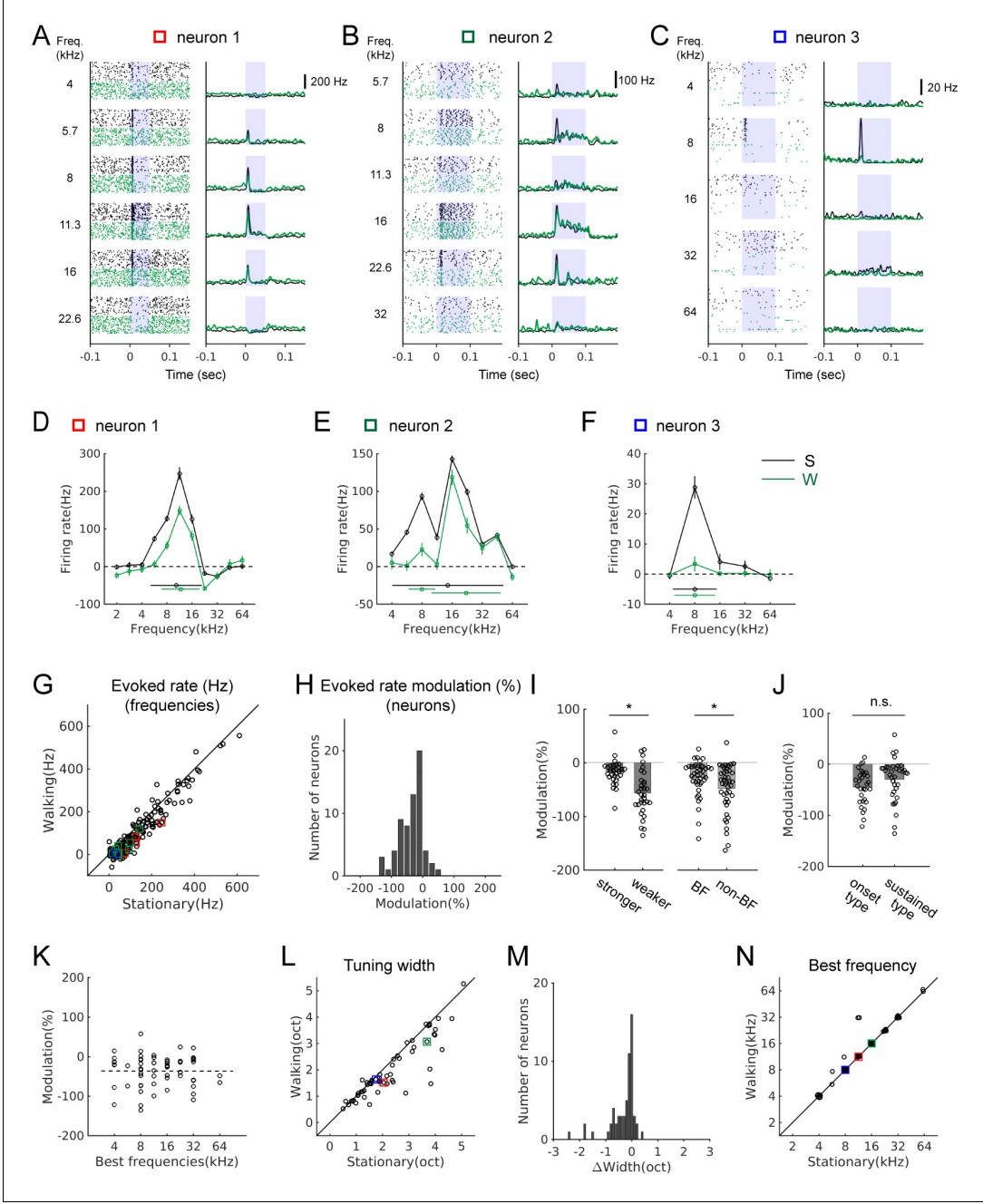

**Figure 4.** Sound-evoked activity of IC neurons is attenuated during locomotion. (**A-C**) Tone-evoked responses from three example IC neurons. Black spike rasters and PSTHs are from stationary trials, and green from walking trials. In each neuron, the same number of trials is shown for each condition (A: 35 trials; B: 25 trials; C: 92 trials). Blue shading indicates the period of sound stimulus presentation. Squares in different colors indicate the data points corresponding to the examples in the population plots in G, L, and N. (**D-F**) Tuning curves (black: stationary, green: walking) based on baseline-subtracted average firing rates at stimulus onset (see Materials and methods). Tuning widths are shown as horizontal bars in each condition (with the centroid shown as a circle at the center). The tuning curves are from the neurons shown in A-C, respectively. (**G**) Comparison of evoked rates between the stationary and walking conditions at all frequencies with excitatory responses (178 tones, 65 neurons). Values from the example neurons in A-C are indicated as squares in corresponding colors. (**H**) Histogram of percent evoked-rate modulation across the recorded neurons (n = 65). One-sample t-test for the zero mean, p=1.4×10⁻¹⁰. (**I**) Comparison of the percent evoked-rate modulation between neurons with weaker and stronger responses (the left two bars; stronger: above the median, n = 33; weaker: below the median, n = 32; t-test, *Figure 4 continued on next page*

*Figure 4 continued*

p=0.0175), or between best and non-best frequencies (BF vs. non-BF) (the right two bars; n = 43 of 65 neurons that had responses at both best and non-best frequencies; paired t-test, p=6.98×10$^{-4}$). Each data point denotes a neuron. (J) Percent evoked-rate modulation in neurons with responses only at the onset vs. neurons with sustained responses (onset type: n = 30; sustained type: n = 35; t-test, p=0.103). (K) Scatter plot of percent evoked-rate modulation as a function of best frequencies (n = 65; r = −0.0031, p=0.98). (L) Frequency tuning widths in octaves for stationary and walking conditions. (M) Histogram of tuning width changes (n = 60, one sample Wilcoxon signed rank test, p=3.0×10$^{-7}$; mean change = −0.29 octaves). (N) Comparison of best frequencies (n = 60). In L-N, neurons that lost all excitatory responses in the walking condition, yielding the tuning width of 0, were not included (n = 5).

The online version of this article includes the following source data and figure supplement(s) for figure 4:

**Source data 1.** Source data for modulation of sound-evoked responses during locomotion, relationships between spontaneous and evoked activity modulation, and spontaneous activity modulation in different response types.
**Figure supplement 1.** Relationships between the locomotion-related modulation of spontaneous and tone-evoked activity.
**Figure supplement 2.** Spontaneous activity modulation during locomotion in neurons with different evoked response patterns.

## Discussion

By recording neural activity in the IC of behaving mice, we found that locomotion can modulate spontaneous activity bidirectionally and attenuates sound-evoked activity. Our results indicate that locomotion-related neural signals are robust and wide-spread at this relatively early stage of the auditory pathway. The prevalence of movement-related signals suggests that multi-modal integration of movement and sound is an essential part of sound processing in the IC.

We observed clear modulation of IC spontaneous activity during locomotion. However, in the MGB, the main target of the IC, prior studies reported no such changes in spontaneous activity (*Zhou et al., 2014*; *Williamson et al., 2015*; *McGinley et al., 2015*). One plausible explanation for this discrepancy is that the modulation of IC spiking activity primarily evokes subthreshold responses in thalamic neurons, making it difficult to detect, but might still modulate sound-evoked activity (*Jain and Shore, 2006*) in the MGB during locomotion (*Williamson et al., 2015*; *McGinley et al., 2015*).

While spontaneous activity was modulated bidirectionally during locomotion, evoked activity was attenuated in most of our recorded IC neurons. The attenuation observed in the IC could explain in part the similar attenuation observed in the MGB neurons by prior studies. In addition, the attenuation appears global because it occurred regardless of the best frequencies or the temporal response patterns. Interestingly, the attenuation of evoked activity led to a decrease in frequency tuning widths, indicating increased neural selectivity for sound frequency (*Figure 4L and M*). The average decrease across the population (~0.3 octaves) was greater than a behaviorally discriminable difference (≤10% or ~0.14–0.15 octaves; *Günter, 1975*; *Clause et al., 2011*; *de Hoz and Nelken, 2014*; *Guo et al., 2017*). Thus, while the attenuation of sound-evoked activity may contribute to poorer tone detection during locomotion (*McGinley et al., 2015*; *Schneider et al., 2018*), the increase in frequency selectivity may improve frequency discrimination (*Aizenberg et al., 2015*; *Carcea et al., 2017*; *Guo et al., 2017*).

It is likely that auditory reafference contributes to the observed modulation, and our results do not exclude contributions of peripheral mechanisms such as bone conduction. However, the negative modulation latency (~100 msec) and the robust modulation in hearing-impaired mice suggest that central mechanisms play the major role. One obvious candidate is the somatosensory inputs to the IC (*Cooper and Young, 1976*; *Aitkin et al., 1978*; *Coleman and Clerici, 1987*; *Zhou and Shore, 2006*; *Lesicko et al., 2016*; *Olthof et al., 2019*), which may provide feedback during locomotion and modulate sound-evoked activity (*Aitkin et al., 1978*; *Jain and Shore, 2006*). Somatosensory projections predominantly target the lateral cortex of the IC (*Lesicko et al., 2016*), but we found modulated neurons throughout the IC (*Figure 1E*). This may occur via connections within the IC (*Rockel and Jones, 1973*; *Coleman and Clerici, 1987*; *Jen et al., 2001*), via multi-sensory inputs from the dorsal cochlear nucleus (*Kanold and Young, 2001*; *Singla et al., 2017*; *Goyer et al., 2019*), or via descending inputs from the somatosensory cortex (*Olthof et al., 2019*). Regardless of

the precise neural mechanisms, our results suggest that somatosensory signals are more widespread across the IC than has been previously hypothesized, and a crucial function of these inputs is providing feedback about ongoing movement.

Our results are also compatible with efference copy of motor signals, or movement-related neuromodulatory signals, described in A1 (*Nelson et al., 2013*; *Schneider et al., 2014*; *Nelson and Mooney, 2016*; *Reimer et al., 2016*). The IC may receive motor-related signals via the descending projections from the motor cortex (*Olthof et al., 2019*). Cholinergic inputs from the peduncular pontine nucleus (*Farley et al., 1983*; *Motts and Schofield, 2009*), which is part of the midbrain locomotion region (*Lee et al., 2014*; *Caggiano et al., 2018*), and noradrenergic inputs from the locus coeruleus (*Klepper and Herbert, 1991*; *Hormigo et al., 2012*), which are active during locomotion (*Reimer et al., 2016*), may also play a role.

Our results indicate that modulation can differ depending on the response types (*Figure 4—figure supplement 2*). In addition to identifying the sources of modulation, it will be important to determine how these signals modulate different cell types such as excitatory, inhibitory (*Ono et al., 2017*), or other specific subtypes (*Oliver et al., 1994*; *Ito et al., 2009*; *Chen et al., 2018*; *Goyer et al., 2019*; *Schofield and Beebe, 2019*).

Neural signals induced by locomotion would inform IC neurons of movement and posture. These movement-related information could then be used for distinguishing self-generated sounds from external ones. Suppression of auditory responses during motor behaviors (*Creutzfeldt et al., 1989*; *Eliades and Wang, 2013*; *Schneider et al., 2014*; *Singla et al., 2017*) is hypothesized to prevent desensitization (*Poulet and Hedwig, 2002*; *Schneider and Mooney, 2018*). This may be one of the functions of the attenuation we observed, especially in environments where walking makes louder sounds (*Schneider et al., 2018*). In addition, movement-related signals in the IC may also enable rapid behavioral responses to a sound source by integrating spatial and body information (*Huffman and Henson, 1990*; *Xiong et al., 2015*). In conclusion, our results suggest that functions of the IC during behavior are highly multi-modal and integrate sound processing with ongoing behavior.

# Materials and methods

**Key resources table**

| Reagent type (species) or resource | Designation | Source or reference | Identifiers | Additional information |
|---|---|---|---|---|
| Strain, strain background (*Mus musculus*) | C57BL/6N | Orient Bio (Korea) | | |
| Genetic reagent (*Mus musculus*) | VGAT-ChR2-EYFP | The Jackson Laboratory | RRID:IMSR_JAX:0145498 | |
| Software, algorithm | MATLAB | MathWorks | RRID:SCR_001622 | All data analysis |
| Software, algorithm | Open Ephys GUI | Open Ephys | | Neural data acquisition |

## Animals

All experiments performed were approved by the Institutional Animal Care and Use Committee of Sungkyunkwan University in accordance with the National Institutes of Health guidelines. Neural recordings were performed in C57BL/6N mice (n = 15) or a transgenic mouse line (VGAT-ChR2-EYFP line with C57BL/6J background in which ChR2-EYFP was targeted to the Slc32a1 gene that encodes VGAT, Jackson Laboratory stock #014548; n = 10) of both sexes, aged 6–10 weeks. Mice were put under a reversed 12 hr light/12 hr dark cycle. Neural recordings were made during the dark cycle. Food and water were accessible at all times. In the present study, we did not use transgene expression in the transgenic line. No differences in the degree of modulation in spontaneous or evoked activity between the two strains were found.

## Headpost surgery

For neural recordings in a head-fixed preparation, a custom metal headpost was cemented to the skull. Mice were positioned in a stereotaxic device (Kopf Instrument, Itujunga, CA) and anesthetized

using Isoflurane (1–4%) delivered via a vaporizer (DRE Veterinary, Louisville, KY). An eye ointment was applied to keep the eyes from drying. A small amount of lidocaine (2%) was injected under the skin overlying the skull, and an incision was made to expose the skull. The connective tissues were gently removed, and the skull surface was allowed to dry. The head was positioned such that difference in the dorso-ventral coordinates of the bregma and lambda was ≤100 μm. A small ground screw was cemented toward the rostral end of the exposed skull surface. For future craniotomy over the IC, markings were made 5 mm posterior to the bregma. A metal headpost was positioned not to obstruct future access to the IC and was secured using dental cement (Super-bond C&B, Sun Medical, Moriyama city, Japan).

## Neurophysiology

Mice were acclimated to head-fixing and walking on a passive circular treadmill for 2–4 days (one 30 min session per day) prior to neurophysiological recordings. On the day of neural recording, a cranial window (~1.8 mm x 1 mm) was made over the IC under isofluorane anesthesia (1–4%). The window was covered with Kwik-Cast (WPI, Sarasota, FL) and mice were allowed to recover for at least 2 hr. Recordings were made from the IC (AP ~5.0 mm posterior to the bregma, ML ~0.4–1.8 mm from the midline) using either a single tungsten electrode or a linear array of tungsten electrodes (~5 MΩ, FHC, Bowdoin, ME). The electrodes were controlled by a single-axis motorized micro-manipulator (IVM Mini, Scientifica, Uckfield, UK). Neural signals were acquired using a 16-channel headstage (RHD2132, Intan Technologies, Los Angeles, CA) and Open Ephys data acquisition hardware and software. Spiking activity was band-pass filtered (600–6000 Hz) and digitized at 30 kHz. Locomotion on a treadmill was detected using a rotary encoder (Scitech Korea, Seoul, Korea), and the output voltage signal was recorded as analog and digital inputs for further analysis.

## Sound stimuli

Pure tone stimuli with a sampling rate of 400 kHz were generated in MATLAB (MathWorks, Natick, MA), and presented at 70 dB SPL using a D/A converter (PCIe-6343, National Instruments), a power amplifier (#70103, Avisoft, Glienicke/Nordbahn, Germany), and an ultrasonic speaker (Vifa, Avisoft). The sound system was periodically calibrated for each tone stimulus frequency using a ¼" microphone (Bruel and Kjaer 4939, Nærum, Denmark). During neurophysiological recordings, the speaker was placed 15 cm from the animal's right ear at 45˚ from the body midline. In initial recordings, the tone stimulus set consisted of frequencies between 4 kHz and 64 kHz in one octave steps (duration: 100 msec; on and off ramps: 1 msec; presented pseudorandomly at 2 Hz). In subsequent experiments, to better estimate frequency tuning, tone stimuli were presented at frequencies between 2 kHz and 64 kHz in half octave steps (duration: 50 msec; on and off ramps: 1 msec; presented pseudorandomly at 4 Hz). Each tone stimulus was repeated at least 20 times, but typically much more (mean: 131 trials; range: 20–320). Broadband noise stimuli (2–64 kHz) for assessing hearing thresholds were 50 msec long (5 msec on and off ramps) and presented at 20–90 dB SPL in 10 dB steps in a pseudorandom order. Each sound level was repeated at least 50 times.

Sounds generated by locomotion on our treadmill were recorded by placing a microphone (CM16/CMPA, Avisoft) close to the legs (*Figure 1—figure supplement 1A*). The recorded sound signal was bandpass filtered (1–25 kHz) and subjected to a noise reduction procedure to reduce the baseline device noise using Audition (Adobe, San Jose, CA). Without noise reduction, the baseline can evoke auditory responses. The sound level of walking sounds was estimated by comparing recorded walking sounds with a series of recorded playbacks at different levels (20 to 40 dB in 5 dB steps). Root mean square (RMS) values were calculated either from a whole representative 2 sec trace or from thresholded walk-related events. To detect events, sound recording was smoothed (two msec rectangular window) and thresholded. Events were first determined from a 40 dB playback recording (threshold = 4% of the peak of the smoothed recording), and the same events were used to compare walking sound recording and playback recordings. Recorded playback at 30 dB had an RMS value close to that of the recorded walking sounds (*Figure 1—figure supplement 1B and C*). Due to the noise reduction procedure, our playback stimulus had fewer smaller events than the recording of walking sounds (*Figure 1—figure supplement 1B*), but sound level comparisons without noise reduction yielded the same results, and the smaller events were much lower in intensity (~8 dB) that the contribution of these events should be small. The playback stimulus was

presented at 30 dB and 40 dB and repeated at least 20 times. At 40 dB, in 5 of 25 neurons, baseline periods evoked substantial auditory responses even with noise reduction, obscuring playback responses. An example is shown in *Figure 1—figure supplement 1E* and these neurons were excluded from group analysis.

Sound stimuli were presented using a custom-written stimulus presenter program written in Python 2.7 (by Jeff Knowles; https://bitbucket.org/spikeCoder/kranky), which communicated with Open Ephys GUI software (http://www.open-ephys.org/gui).

## Deafening

To prevent mice from hearing sounds generated by locomotion, a bilateral deafening procedure was performed. Mice were anesthetized by intraperitoneal injection of Ketamine/Xylazine (100 mg/ 10 mg per kg). A small incision was made ventral and posterior to the pinna. To expose the auditory bulla that surrounds the middle ear cavity, the overlying tissue was gently spread using fine forceps. An opening was made in the bulla to visualize the ossicles and cochlea. The ossicles were removed and kanamycin drops (1 mg/ml) were applied 3–4 times at the oval window of the cochlea in an attempt to induce further hair cell damage. The middle ear cavity was filled with gelfoam, the overlying tissue was closed, and the skin was sutured. Mice were given analgesics (meloxicam, 5 mg/kg) and allowed to recover for 2–3 days before receiving headpost surgery for neural recording.

## Data analysis

Spikes were detected and sorted offline using commercial spike sorting software (Offline Sorter v4, Plexon, Dallas, TX). Detected spike waveforms were clustered using principal component analysis (PCA), and clusters with a clear separation in PC space were taken as single units. A refractory period of 0.7 msec was imposed, and the rate of refractory period violation was required to be $\leq 0.5\%$.

Spontaneous activity was measured either from the baseline period that preceded a tone presentation (0.1 or 0.2 sec) or from 1 sec segments of a continuous recording of spontaneous activity (>200 sec). Periods of locomotion were determined by thresholding the speed of the treadmill at 2 cm/sec, obtained by smoothing the digital recordings of the treadmill sensor output (200 msec hanning filter; a transition from low to high, or vice versa, corresponded to 0.26 cm). During the stationary periods, occasional short blips of movement occurred, but otherwise, the speed was 0. The segments of spontaneous activity were assigned to either a stationary or a walking condition, and only those that occurred entirely during one of the behavioral conditions were included for analysis.

To analyze the timing of neural modulation relative to movement onset (*Figure 2*), walking periods with clear onsets were identified. Then, for each walking period, the onset was defined as the time when the treadmill sensor output changed by 2% (corresponding to <1 mm of travel) of the maximum range (2.5V). Once locomotion onsets (=onset(L)) were defined, onset-triggered averaging of the smoothed firing rate was performed. The onset of neural activity modulation (=onset(M)) was defined as the time when the 95% confidence interval of the onset(L)-triggered firing rate first deviated from the average stationary rate by two times the standard deviation. The confidence intervals were obtained using bootstrap resampling of the onset(L)-triggered firing rate segments. Latency of modulation was defined as the time of onset(M) - the time of onset(L). Modulation onset latency could be determined only in neurons that yielded a clear onset(L)-triggered average (normal: n = 30 of 73 modulated neurons; hearing impaired: n = 14 of 23 modulated neurons). The latency could not be determined in the remaining neurons due to insufficient locomotion onsets or relatively weak modulation.

To assess the effect of deafening, multi-unit responses to broadband noise (2–64 kHz) were examined across the IC for a range of sound levels in 10 dB steps (20–90 dB). Multi-unit spike times were obtained by thresholding the neural recordings at three times the standard deviation of the baseline noise. Rate-level curves were constructed based on the peak evoked firing rates (*Figure 3A and B*). Rate-level curves were obtained from a few tens of multi-unit sites for each deafened mouse (4 mice, 43 ± 13 sites per mouse).

To compare the degrees of modulation between the normal hearing and hearing-impaired mice (*Figure 3—figure supplement 1*) and among response types (*Figure 4—figure supplement 2*), a modulation index was used: MI = [<r>(walk) – <r>(stationary)] / [<r>(walk) + <r>(stationary)],

where <r> represents the average firing rate and MI values varied between −1 and +1 (*Rummell et al., 2016*). Because deafening decreased spontaneous firing rates (21.9 ± 2.2 Hz vs. 14.3 ± 1.8 Hz), the MI allowed us to compare the relative changes induced by locomotion.

Tone-evoked activity was analyzed in neurons with excitatory response to at least one of the presented tone frequencies. A significant response to a tone stimulus was determined using a paired t-test between the firing rate during the baseline period preceding the tone and that during a response window. A response window was defined around the time of the peak of the smoothed peristimulus time histogram (PSTH) (from 5 msec before and to 7 msec after the peak; smoothing by a Gaussian function with the standard deviation of 2 msec) at a neuron's best frequency (the tone frequency with the greatest peak response). The same response window was used for all stimuli in a given neuron. Tone-evoked activity was quantified as the average firing rate during a response window minus the spontaneous firing rate. Spontaneous firing rate was measured during a 100 or 200 msec period preceding each stimulus presentation. Evoked trials were assigned to either a stationary or a walking condition as in the analysis of spontaneous activity described above. Only neurons that had at least five repeats for a tone stimulus in both conditions were included (Stationary: 131 ± 47 trials per stimulus; Walking: 40 ± 26 trials per stimulus). Modulation of evoked activity was quantified as percent change in evoked rate (100*[r_evoked(walking) − r_evoked(stationary)] / [r_evoked(stationary)]). Percent change in the evoked rate for a neuron was defined as percent change summed over all responsive tone frequencies. Modulation analysis using a fixed 15 msec response window starting 5 msec after the sound onset yielded similar results.

Tuning curves were constructed from the average evoked firing rates at different tone frequencies. To quantify tuning widths, tuning curves were first linearly interpolated between neighboring frequencies (100 points), and tuning widths were expressed as four times the second moment about the centroid, measured in octaves (*Escabí et al., 2007*; *Ono et al., 2017*). In rare cases (n = 2) where multiple tuning width segments occurred due to a non-responsive tone in the middle, the widths were summed minus the overlap.

## Histology

At the end of a recording session, small lesions were made by applying current (30 µA, 10 sec) through the recording electrodes. Animals were then transcardially perfused using phosphate-buffered saline (PBS) followed by 4% paraformaldehyde. Brains were post-fixed for at least a day and cryoprotected in 30% sucrose before they were cut on a cryostat. Sections were cut at 40 µm thickness, mounted on slides, and processed for Nissl staining. Recording locations were estimated based on the locations of the lesion.

## Statistical analysis

All statistical analysis was performed in MATLAB. The significance level was $\alpha$ = 0.05 except for spontaneous activity modulation, where $\alpha$ = 0.01. Normality was assessed using Lilliefors test (lilliettest function in MATLAB), and when data significantly deviated from normal distribution, nonparametric tests were used. Results are presented as mean ± SEM unless otherwise noted.

Statistical significance of the modulation in spontaneous firing was determined using the mean firing rates from recording segments (see *Data analysis* above) during which no sound stimuli were presented. The spontaneous firing rates of the analyzed segments were generally not normally distributed, so the significance of the modulation was determined using a permutation test. In each permutation, segments of spontaneous activity from the stationary or walking groups were combined and then randomly assigned to two groups with the original sample sizes. A distribution of the differences of the means between the two groups was obtained from 1000 permutations. A two-tailed *p* value was obtained by calculating the probability that permuted differences were more extreme than the sample mean difference. For each neuron, p<0.01 was considered a significant modulation. Based on this analysis, neurons were categorized into three groups as those with increased spontaneous activity, those with decreased spontaneous activity, and those with no significant change (normal: n = 96 neurons from 21 mice; hearing impaired: n = 34 neurons from 4 mice; *Figures 1C–E* and *3E–G*, *Figure 3—figure supplement 1A and B*, *Figure 4—figure supplement 1E–H*, and *Figure 4—figure supplement 1A–C*).

For the analysis of walking sound playback (*Figure 1—figure supplement 1*), only stationary trials were used (17 ± 5 trials). In *Figure 1—figure supplement 1F and G*, t-tests were performed to determine whether the correlation coefficients between locomotion- and playback-induced firing rate changes were significantly different from 0 (n = 25 at 30 dB; n = 20 at 40 dB).

In *Figure 2D*, a Wilcoxon rank sum test was performed to determine if median latencies of neural modulation relative to locomotion onset differed significantly between the neurons with increased (n = 22) and decreased (n = 8) spontaneous firing rates during locomotion. Similarly, in *Figure 3—figure supplement 1C*, latencies were compared between normal (n = 30) and hearing-impaired mice (n = 14). In *Figure 4—figure supplement 2D*, the Kruskal-Wallis test was used (onset: n = 5; sustained: n = 13; suppressed: n = 8).

In *Figure 3—figure supplement 1B*, to determine whether the degrees of modulation differed significantly between the normal and hearing-impaired mice, t-tests were performed on MI values in the neurons with increased (*Figure 3—figure supplement 1B*, left; normal: n = 51 of 96 neurons; hearing impaired: n = 16 of 34 neurons) or decreased spontaneous firing (*Figure 3—figure supplement 1B*, right; normal: n = 22 of 96 neurons; hearing impaired: n = 7 of 34 neurons). In *Figure 4—figure supplement 2B and C*, one-way ANOVA was used to compare MI values across the three response types. Statistical comparisons using percent changes in firing rate yielded similar results.

For modulation of evoked activity (*Figure 4*), of the 67 neurons in which tone stimuli evoked excitatory responses, 65 were included in the analysis. Two neurons had unreliable percent change values due to evoked rates close to 0 and were excluded from the analysis. In *Figure 4H*, a one sample t-test was performed to determine whether the population mean of percent changes in evoked rates was significantly different from zero (n = 65). In *Figure 4I* (two bars on the left), the 65 neurons were divided into two groups: those with relatively stronger responses (above the median, n = 33) and those with weaker responses (below the median, n = 32). Then, a two-sample t-test was performed to determine whether the evoked rate change differed between the two groups. In *Figure 4I* (two bars on the right), percent changes in the evoked rate were compared between the best and non-best frequencies of a neuron using a paired t-test (n = 43 of 65 neurons with multiple frequencies with response). In *Figure 4J*, to determine whether percent evoked rate change differed depending on response type, a two-sample t-test was performed between neurons with onset responses only (n = 30 neurons) vs. those with onset followed by sustained responses (n = 35 neurons). In *Figure 4K*, to determine whether the correlation coefficient between the best frequencies and changes in evoked rate was significantly different from 0, a t-test was used (n = 65). In *Figure 4M*, to determine whether the population median frequency tuning width change was significantly different from 0, a one-sample Wilcoxon signed rank test was used (n = 60). In *Figure 4L–N*, neurons that lost all excitatory responses during locomotion, yielding the tuning width of 0, were excluded (n = 5).

In *Figure 4—figure supplement 1E*, percent changes in evoked rate were compared across the three different spontaneous activity modulation categories using a one-way ANOVA (of 65 neurons analyzed for evoked response, 41 showed increased, 7 showed decreased, and 17 showed no change in spontaneous activity). In *Figure 4—figure supplement 1F–H*, to determine whether the correlation coefficient between the modulation of spontaneous and evoked rate was significantly different from 0, t-tests were performed for the increased (n = 41), decreased (n = 7), or no change (n = 17) spontaneous modulation groups.

# Acknowledgements

This study was supported by the Institute for Basic Science in Korea (IBS-R015-D1). We thank Hyesook Lee for her help with histology, Dr. Seong-Gi Kim for helpful discussions and support, Drs. Oliver James, Karl Kandler, and Mimi Kao for their critical comments on earlier versions of the manuscript.

## Additional information

### Funding

| Funder | Grant reference number | Author |
|---|---|---|
| Institute for Basic Science | IBS-R015-D1 | Joonyeol Lee<br>Gunsoo Kim |

The funders had no role in study design, data collection and interpretation, or the decision to submit the work for publication.

### Author contributions

Yoonsun Yang, Conceptualization, Formal analysis, Investigation, Methodology, Writing – Original Draft Preparation, Writing – Review and Editing; Joonyeol Lee, Resources, Funding acquisition, Methodology; Gunsoo Kim, Conceptualization, Formal analysis, Funding acquisition, Investigation, Methodology, Writing – Original Draft Preparation, Writing – Review and Editing

### Author ORCIDs

Joonyeol Lee (iD) https://orcid.org/0000-0001-9929-6080
Gunsoo Kim (iD) https://orcid.org/0000-0001-9318-8329

### Ethics

Animal experimentation: This study was performed in strict accordance with the recommendations in the Guide for the Care and Use of Laboratory Animals of the National Institutes of Health. All of the animals were handled according to the protocol (SKKUIACUC2018-02-09-1) approved by the institutional animal care and use committee (IACUC) of the Sungkyunkwan University. Surgeries were performed under isofluorane or ketamine/xylazine anesthesia, and every effort was made to minimize suffering.

### Decision letter and Author response

Decision letter https://doi.org/10.7554/eLife.52228.sa1
Author response https://doi.org/10.7554/eLife.52228.sa2

## Additional files

### Supplementary files

• Transparent reporting form

### Data availability

All data generated or analysed during this study are included in the manuscript and supporting files. Source data files have been provided for Figures 1 through 4.

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
