## [Decision Letter]

**Acceptance summary:**

This study is a clear observation that both spontaneous and evoked activity in the inferior colliculus, a crucial subcortical stage of the auditory system, is modulated by locomotion in mice. While several reports already indicate that auditory cortex is impacted by locomotion, it is striking to observe that such modulations exist earlier in the auditory system. This suggests that the auditory colliculus, like its visual counterpart is a site of sensory-motor integration. What are the mechanisms and the role of this early integration? This remains to be precisely addressed in future research in order to decipher the purpose of the multiplicity of parallel sensory-motor loops in the brain.

**Decision letter after peer review:**

Thank you for submitting your article "Integration of locomotion and auditory signals in the mouse inferior colliculus" for consideration by *eLife*. Your article has been reviewed by three peer reviewers, including Brice Bathellier as the Reviewing Editor and Reviewer #1, and the evaluation has been overseen by Barbara Shinn-Cunningham as the Senior Editor. The following individual involved in review of your submission has agreed to reveal their identity: Ursula Koch (Reviewer #3).

The reviewers have discussed the reviews with one another and the Reviewing Editor has drafted this decision to help you prepare a revised submission.

Summary:

The authors make extracellular recordings of single unit activity in the inferior colliculus from head-fixed mice on a treadmill and demonstrate that head-fixed locomotion modulates inferior colliculus activity. In particular, the authors show that, in a large proportion of IC neurons, walking modulates spontaneous activity either up or down. In contrast sound evoked activity levels are mainly attenuated by movement, similarly to the effects observed in the MGB and the auditory cortex. Also neurons that show movement-related activity changes are located throughout the inferior colliculus. This is especially interesting in the light of a recent publication by Olthof et al., 2019 which demonstrates widespread projections from non-auditory regions, including motor regions to the entire inferior colliculus, whereas previous studies had suggested that mainly dorsal and external cortex of the IC were activated by other sensory modalities.

Suppression of sound-evoked activity during movement has been noted in the cortex, thalamus and cochlear nucleus, though not always using a treadmill. The reviewers agree that data acquisition and analysis are solid, especially for the analysis of movement related effects on sound evoked activity and a number of control experiments to exclude self-generated auditory stimulation by movements. The main finding is convincing and the graphical presentation of the data is compelling. However, all in all, this is a simple, descriptive study. The controls with passive playback are nice but are perhaps better suited as a supplemental figure as they are not fully conclusive. The controls with deafened animals are important but some details also could be shown in a supplementary figure. For these reasons the reviewers agree that the article's main findings should be communicated in a more compact manner. We would like to consider this article for publication however in the format of a short report https://reviewer.elifesciences.org/author-guide/types. Also, the following comments should be addressed in the resubmission.

Essential revisions:

1) Playback experiments are not compelling and may not suffice in its current form to exclude that the increase in spontaneous activity by walking is mainly an effect by the sound produced by walking on the treadmill.

The assumption is that a microphone aimed at the legs is capturing the signal reaching the inner ear. It is entirely possible that it does not. As a simple example, jiggle your earlobe. It generates quite an intense sound in the ear canal that would not be measurable elsewhere. Beyond this, sound can be transmitted by bone conduction. These caveats should be mentioned in the Discussion although I think the hearing loss control removes any real concern about the conclusion.

Hence the authors should also rather put this analysis as a supplementary figure where they could provide more details on the recoding methods and on the analysis of the recorded sounds and how variations in this sound affect spiking activity in these neurons.

2) The deafening experiment provide further evidence that movement related spontaneous activity changes are not induced by sound activation. There is one caveat in the interpretation of the data. The fact that the delay between deafening and recording maybe as long as 7 days may initiate synaptic reorganization. This could be discussed.

3) The authors do not correlate modulation in spontaneous activity with firing pattern. This may indicate whether movement differentially affects different neuron types in the IC. Could the author comment on this or provide further analysis?

4) The analyses are carefully presented and figures are well executed, but there is a concern with Figure 7G which plots correlation between baseline modulation and modulation of sound-evoked response. However, the sound evoked responses are not baseline subtracted here, which as the author show in Figure 7F diminishes strongly the impact of locomotion of sound responses. I think that a fairer plot would be to use baseline subtracted response. Also the plot and the correlation measures should be computed with baseline subtracted response for all cell groups (excitatory, inhibitory, no modulation on baseline), and for each groups individually. In general, non-baseline subtracted measures could be removed or put in a supplementary figure.

[Editors' note: further revisions were suggested prior to acceptance, as described below.]

Thank you for resubmitting your work entitled "Integration of locomotion and auditory signals in the mouse inferior colliculus" for further consideration by *eLife*. Your revised article has been evaluated by Barbara Shinn-Cunningham (Senior Editor) and a Reviewing Editor.

The manuscript has been improved but there are some remaining issues that need to be addressed before acceptance, as outlined below:

The revisions are overall satisfactory, however there is still an issue with the quantification of the walking sound playback experiment in Figure 1—figure supplement 1F and G. The authors compare IC responses during walking and responses to sounds emitted by walking at two intensity. However this comparison is made for different categories of responses (up, down modulations and responses to playback much larger than to walking). This is too much cherry picking and this probably prevents seeing that simply there is little correlation between walking and playback sound responses. Moreover the paragraph in the results describing this observation is very confusing and reports differences between the two conditions that go in all directions. For clarity, it would be much better if the authors just plot the responses during walking against response for playback (one plot for playback at 30dB one for playback at 40dB), measure the correlation and conclude based on the strength of the correlation. Eventually (depending on correlation value), it should be said that some part of the responses seen during walking may be due to the sound produced by the animal.

---

## [Author Response]

Essential revisions:1) Playback experiments are not compelling and may not suffice in its current form to exclude that the increase in spontaneous activity by walking is mainly an effect by the sound produced by walking on the treadmill.The assumption is that a microphone aimed at the legs is capturing the signal reaching the inner ear. It is entirely possible that it does not. As a simple example, jiggle your earlobe. It generates quite an intense sound in the ear canal that would not be measurable elsewhere. Beyond this, sound can be transmitted by bone conduction. These caveats should be mentioned in the Discussion although I think the hearing loss control removes any real concern about the conclusion.Hence the authors should also rather put this analysis as a supplementary figure where they could provide more details on the recoding methods and on the analysis of the recorded sounds and how variations in this sound affect spiking activity in these neurons.

It is difficult to know the perceived intensity of walking sounds, and we share the reviewers’ concern that we may be underestimating the contribution of walking sounds. As suggested, we now present the playback results as a supplementary figure (Figure 1—figure supplement 1). We describe in detail how we estimated the sound level of the recorded walking sounds in the Materials and methods section and in Figure 1—figure supplement 1A-C. We also present the results from the walking sound playback at 40 dB in Figure 1—figure supplement 1D-G. We have removed the old Figure 3B because this example response plot is redundant. We have also removed the old Figure 3C that showed the overall firing rate change distributions because the within neuron comparisons presented are sufficient and more meaningful.

In Figure 1—figure supplement 1A-C, we show how we compared the recording of walking sounds with the recording of the playbacks. This analysis is also described in the Materials and methods section:

“Sounds generated by locomotion on our treadmill were recorded by placing a microphone (CM16/CMPA, Avisoft) close to the legs (Figure 1—figure supplement 1A). […]An example is shown in Figure 1—figure supplement 1E and these neurons were excluded from group analysis.”

In Figure 1—figure supplement 1D-G, we now present results with the playback at 40 dB – a much higher sound level. At 40 dB, playback responses increased substantially. Therefore, assuming no suppressive mechanisms for self-generated sounds (Singla et al., 2017; Schneider et al., 2018), if the perceived walking sounds are louder than our estimation, the contribution of auditory reafference will increase. This is now acknowledged in the Results section:

“When mice walk on a treadmill, their movements generate low-intensity sounds. These sounds may modulate IC neuron activity. […] However, stronger playback responses at 40 dB suggest that if the perceived intensity of walking sounds is higher than estimated, the contribution of auditory reafference could be higher (Figure 1—figure supplement 1D-G).”

We also mention that auditory reafference and bone conduction could contribute to the observed modulation in the Discussion:

“It is likely that auditory reafference contributes to the observed modulation, and our results do not exclude contributions of peripheral mechanisms such as bone conduction, However, the negative modulation latency (~100 msec) and the robust modulation in hearing-impaired mice suggest that central mechanisms play the major role.”

2) The deafening experiment provide further evidence that movement related spontaneous activity changes are not induced by sound activation. There is one caveat in the interpretation of the data. The fact that the delay between deafening and recording maybe as long as 7 days may initiate synaptic reorganization. This could be discussed.

We agree with the reviewers that during the period between deafening and recording, hearing loss can induce reorganizations in the IC. Our results do not exclude that this plasticity could influence movement-related modulation. However, in our deafening experiments, movement-related modulation was very similar between normal and hearing-impaired mice – the proportions of the modulation type, the magnitude, and the timing of modulation (Figure 3—figure supplement 1). While there may be quantitative changes in locomotion-related modulation with hearing loss, this observation makes it likely that we are observing the same phenomenon in both groups.

The caveat for deafening experiments is now mentioned in the Results:

“Deafening may induce plasticity in the IC during the period before neural recording (8.2 ± 1.1 days; Chambers et al., 2016). However, not only that locomotion-modulated neurons were found in hearing-impaired mice, but also the modulation was similar to that in normal mice in terms of the types, magnitudes, and latencies (Figure 3—figure supplement 1). Although we cannot exclude the influence of deafening-induced plasticity, the existence of locomotion-modulated neurons in hearing-impaired mice with similar characteristics provides strong evidence that non-auditory signals modulate the activity of IC neurons during locomotion.”

3) The authors do not correlate modulation in spontaneous activity with firing pattern. This may indicate whether movement differentially affects different neuron types in the IC. Could the author comment on this or provide further analysis?

We thank the reviewers for this helpful suggestion. We performed the suggested analysis to see if there was any relationship between the modulation of spontaneous activity and sound-evoked response patterns (onset only, sustained, and suppression only). Our analysis showed that the sustained type had a higher proportion of excitatory modulation (71% vs. 52% and 36%), and the suppressed type had a higher proportion of suppressive modulation (41% vs. 14% and 12%) (Figure 4—figure supplement 2A). Thus, neurons that showed a sustained excitation by sound stimulation tended to become excited during locomotion, and neurons that were only suppressed by sound tended to become suppressed during locomotion. These results suggest that there is an overlap in the inputs for the locomotion-induced and sound-induced modulation. However, the overlap must be only partial because many (36%) neurons that were only suppressed by sounds could be excited during locomotion. We did not find evidence that the degrees of or latencies for modulation differed significantly depending on response type (Figure 4—figure supplement 2B-D).

We describe this analysis and its implications in the Results section:

“We next evaluated whether there is a relationship between the modulation of spontaneous and sound-evoked activity, which might indicate shared neural mechanisms. […] These correlations suggest an overlap between modulatory and sound-driven inputs, but this overlap is partial because the suppressed type can also be excited by locomotion.”

4) The analyses are carefully presented and figures are well executed, but there is a concern with Figure 7G which plots correlation between baseline modulation and modulation of sound-evoked response. However, the sound evoked responses are not baseline subtracted here, which as the author show in Figure 7F diminishes strongly the impact of locomotion of sound responses. I think that a fairer plot would be to use baseline subtracted response. Also the plot and the correlation measures should be computed with baseline subtracted response for all cell groups (excitatory, inhibitory, no modulation on baseline), and for each groups individually. In general, non-baseline subtracted measures could be removed or put in a supplementary figure.

We have removed Figure 7F that showed firing rates without baseline subtraction. We now also present three separate correlation plots with percent changes in spontaneous and evoked rates, for the positive, negative, and no modulation groups. The modulation index is no longer used for this analysis because evoked rates can be negative, which prevents constructing the same index (Figure 4—figure supplement 1F-H).

During the analysis for these correlation plots, we discovered errors in the assignment of spontaneous modulation types (increased, decreased, and no change). All neurons were re-inspected for significance and direction of modulation, and six neurons changed their categories. Figure 4—figure supplement 1E (old Figure 7E), and 1F-H now reflect the correction. This correction did not affect the conclusions.

[Editors' note: further revisions were suggested prior to acceptance, as described below.]

The revisions are overall satisfactory, however there is still an issue with the quantification of the walking sound playback experiment in Figure 1—figure supplement 1 F and G. The authors compare IC responses during walking and responses to sounds emitted by walking at two intensity. However this comparison is made for different categories of responses (up, down modulations and responses to playback much larger than to walking). This is too much cherry picking and this probably prevents seeing that simply there is little correlation between walking and playback sound responses. Moreover the paragraph in the results describing this observation is very confusing and reports differences between the two conditions that go in all directions. For clarity, it would be much better if the authors just plot the responses during walking against response for playback (one plot for playback at 30dB one for playback at 40dB), measure the correlation and conclude based on the strength of the correlation. Eventually (depending on correlation value), it should be said that some part of the responses seen during walking may be due to the sound produced by the animal.

We thank the editors for this very helpful suggestion. We now plot locomotion-induced firing rate changes against playback responses in Figure 1—figure supplement 1F (30 dB) and G (40 dB). We have removed the old Figure 1—figure supplement 1F and G bar graphs. We also have updated the figure legend and the statistical analysis description in the Materials and methods to reflect the changes. The source data file has also been updated.

The description in the Results now reads:

“When mice walk on a treadmill, their movements generate low-intensity sounds. These sounds may modulate IC neuron activity. We examined this contribution of auditory reafference by playing back recorded walking sounds at a level similar to the recording (30 dB, n = 25; Figure 1—figure supplement 1). We also presented the walking sounds at a substantially higher level (40 dB) to examine the variation in neural response. Playbacks evoked neural responses, but the response magnitudes did not significantly correlate with those induced by locomotion (Figure 1—figure supplement 1D-G). This suggests that although some of the observed modulation may be induced by walking sounds, it is unlikely that the modulation is mainly due to auditory reafference.”